# Assessment and Calibration of a Low-Cost PM$_{2.5}$ Sensor Using Machine Learning (HybridLSTM Neural Network): Feasibility Study to Build an Air Quality Monitoring System

Donggeun Park *,†[ID], Geon-Woo Yoo †, Seong-Ho Park † and Jong-Hyeon Lee †

Research Institute of Environmental Health and Safety, 410, Jeongseojin-ro, Seo-gu, Incheon 404-844, Korea; gw.yoo@ehrnc.com (G.-W.Y.); sh.park@ehrnc.com (S.-H.P.); j.lee@ehrnc.com (J.-H.L.)
* Correspondence: dgpark94@pusan.ac.kr
† These authors contributed equally to this work.

**Abstract:** Commercially available low-cost air quality sensors have low accuracy. The improved accuracy of low-cost PM$_{2.5}$ sensors allows the use of low-cost sensor systems to reasonably investigate PM$_{2.5}$ emissions from industrial activities or to accurately estimate individual exposure to PM$_{2.5}$. In this work, we developed a new PM$_{2.5}$ calibration model (HybridLSTM) by combining a deep neural network (DNN) optimized in calibration problems and a long short-term memory (LSTM) neural network optimized in time-dependent characteristics to improve the performance of conventional calibration algorithms of low-cost PM sensors. The PM$_{2.5}$ concentrations, temperature and humidity by low-cost sensors and gravimetric-based PM$_{2.5}$ measuring instrument were sampled for a sufficiently long time. The proposed model was compared with benchmarks (multiple linear regression model (MLR), DNN model) and low-cost sensor results. The gravimetric measurements were used as reference data to evaluate sensor accuracy. For root-mean-square error (RMSE) for PM$_{2.5}$ concentrations, the proposed model reduced 41–60% of error when compared with the raw data of low-cost sensors, reduced 30–51% of error when compared with the MLR model and reduced 8–40% of error when compared with the MLR model. R$^2$ of HybridLSTM, DNN, MLR and raw data were 93, 90, 80 and 59%, respectively. HybridLSTM showed the state-of-the-art calibration performance for a low-cost PM sensor. In other words, the proposed ML model has state-of-the-art calibration performance among the tested calibration algorithms.

**Keywords:** machine learning; deep learning; calibration; air quality; low-cost sensors; exposure assessment

## 1. Introduction

Air pollution caused by industrialization and urbanization is causing serious environmental and health problems. For example, fine particulate matter (PM) is generated from various emission sources of industrial activities such as industry, transportation and combustion. In particular, fine dust with a diameter of less than 2.5 μm (PM$_{2.5}$) causes various diseases, such as cardiovascular diseases, asthma, and neurotoxicity, because it is directly exposed to the lungs and circulatory system. Therefore, it is very important to obtain the data for regulation on industrial emission by monitoring the PM$_{2.5}$ concentration generated by the emission activity [1].

In South Korea, a gravimetric-based PM$_{2.5}$ measuring instrument has been used as a national reference method (NRM) to monitor the PM$_{2.5}$ concentrations. However, it is expensive to install NRM equipment at the sampling location for each close distance (>USD 10,000) [2]. This limits obtaining PM$_{2.5}$ information from the NRM method at the community level.

Light-scattering low-cost PM$_{2.5}$ sensors are paradigm to solve the cost problem. Since the low-cost sensor can obtain the PM concentration in real-time, it has been used in

various studies such as personal exposure assessment [3,4], indoor exposure estimation [5] and outdoor monitoring [6–8]. However, low-cost sensors are sensitive to environmental variables such as temperature and humidity due to their light scattering method. Badura et al. [9] conducted a validation test to evaluate the reliability of the low-cost sensors in an outdoor field over a long period using the national standard measuring equipment. Above 80% relative humidity, raw data by low-cost sensors observed an apparent overestimation of $PM_{2.5}$ concentration measurements.

Vogt et al. [10] performed the comparison of three models of low-cost $PM_{2.5}$ sensors (Plantower 5003, Sensirion SPS 30 and Alphasense OPC-N3) against the gravimetric device in outdoor field. The SPS 30 sensor has higher accuracy and high correlation compared to other low-cost sensors. However, it has still been shown that the low-cost sensor has lower accuracy than the national standard measurements due to the limitations of the physical characteristics of the sensor.

Zusmana et al. [11] developed metropolitan region-specific calibration models based on the multi-linear regression method (MLR) and the time-series data by various low-cost sensors ($PM_{2.5}$, temperature and humidity) and the NRM network equipment ($PM_{2.5}$) to solve the sensitivity problem driven by environmental variables. The calibration model confirmed the possibility of applying a low-cost sensor at the community level by solving the accuracy degradation caused by the physical characteristics of low-cost sensor. However, the metropolitan region-specific calibration model still showed low accuracy ($R^2$ = 0.67–0.84) in a specific data period.

Si et al. [12] introduced machine learning approaches to improve the accuracy problem of the linear regression method for the low-cost sensor calibration. They compared the $PM_{2.5}$ data calibrated by the simple linear regression (SLR), the multiple linear regression (MLR), the tree-based machine learning algorithm (XGboost) and deep neural networks (DNN) against $PM_{2.5}$ data by the Synchronized Hybrid Ambient Real-time Particulate (SHARP) monitor. They showed the machine learning methods have superior calibration performance compared to the linear regression methods. Among the calibration algorithms, DNN showed the best performance for $PM_{2.5}$ calibrations (person R = 0.85, root-mean-square error = 3.91). However, the calibration performance of low-cost sensors can be still improved because various machine learning algorithms can be fused for the purpose of solving them.

In this study, we propose a state-of-the-art $PM_{2.5}$ calibration model (HybridLSTM) by combining the deep neural network (DNN) optimized in calibration problems and a long short-term memory (LSTM) neural network optimized in time-dependent characteristics to improve the performance of conventional calibration algorithms (DNN, MLR) of a low-cost PM sensor. This work develops a low-cost calibration machine learning (ML) model and compares it with the previous state-of-the-art model (DNN) and conventional MLR model.

The process of this study is shown in Figure 1. First, low-cost Sensirion SPS 30 and NRM equipment were collocated to develop the ML model. If high concentrations of $PM_{2.5}$ are not sampled, incorrect performance evaluation results may be found [11]. Therefore, the experiment is carried out until more than 50 μg/m$^3$ of $PM_{2.5}$ samples are obtained. HybridLSTM, DNN and MLR models are developed based on the obtained data sample (training set), and model performance is compared using the other independent data not used for model development (test set). The MLR and DNN are used as a benchmarks for evaluating the newly developed calibration HybridLSTM model.

The novelty of this study is as follows: improvement of the calibration performance against the calibration model (MLR and DNN) by using a new machine learning algorithm (HybridLSTM). As far as we can tell, this paper is the first report to propose the new approach of calibration of low-cost PM2.5 sensors by using the HybridLSTM algorithm. It is very important because it does not only provide reliable community-level monitoring, but can also help exposure assessment in epidemiological studies.

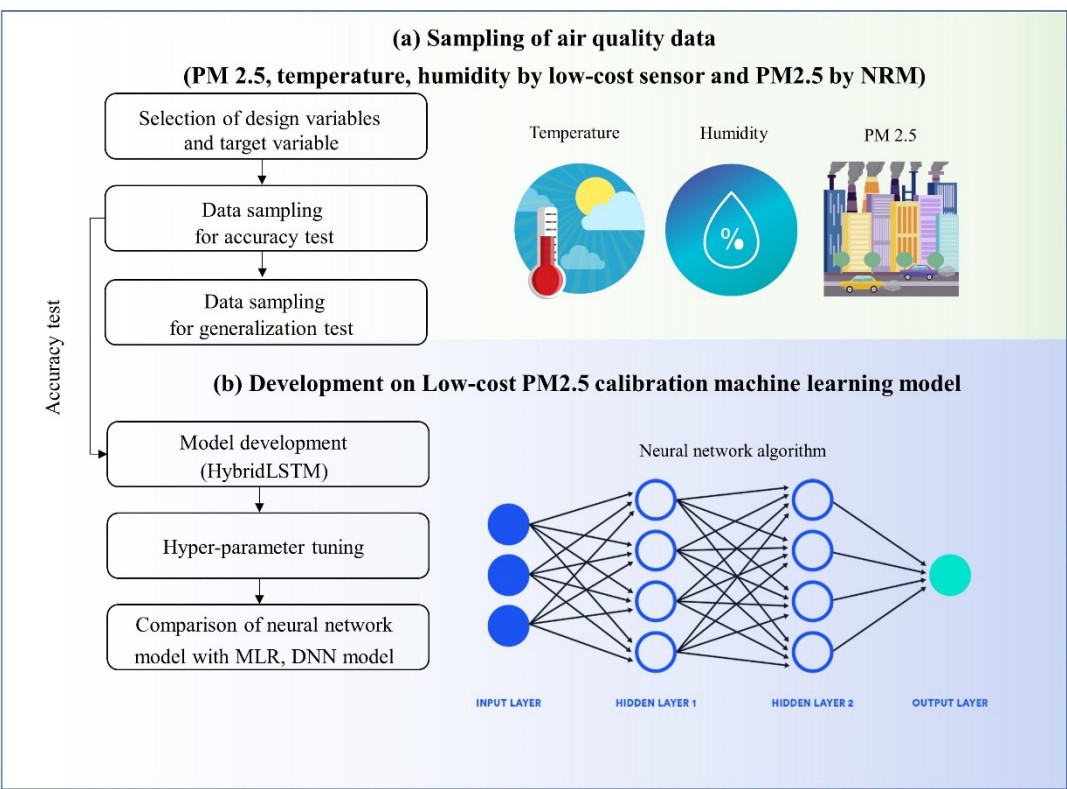

**Figure 1.** Research flow chart. This work is performed in two steps: (**a**) data collection for PM$_{2.5}$, temperature and humidity by low-cost sensor and PM$_{2.5}$ by gravimetric instrument with high accuracy. (**b**) Machine learning model development based on the collected dataset. Calibration performance from the developed model is compared with raw data by low-cost sensor and calibration results by benchmark method (multi-linear regression method, DNN).

## 2. Methods

### 2.1. Data Sampling to Develop Calibration Machine Learning (ML) Model

2.1.1. Air Quality Measurement Instruments

In general, the ML algorithm functions as the relationship between input variables and output variables. In this study, input variables were set as PM$_{2.5}$ by Sensirion low-cost SPS 30 (<USD 50) and temperature and humidity by Sensirion SHT85 (<USD 30) to model the complex relation between environmental variables and PM$_{2.5}$ of the light-scatter method.

It is very important to have consistent precision among low-cost sensors in order to build a monitoring sensor network system by the ML model. Because the low-cost Sensirion SPS 30 has excellent inter-sensor precision with coefficients of determination above 0.9 [10], the ML model based on SPS 30 has the possibility of maintaining consistent performance even with new sensors. Therefore, the PM$_{2.5}$ measurement results by the SPS 30 sensor are set as the input variable based on previous literature [10]. Environmental variables such as temperature and humidity have an effect on decreasing the accuracy of a low-cost sensor based on the light-scatter method [13]. Therefore, two environmental variables were also set as input conditions to model the complex physical characteristics among PM$_{2.5}$, temperature and humidity.

The target variable is PM$_{2.5}$ concentration measured from the gravimetric instrument. The quality of target variable plays an important role in developing the ML model with high calibration performance. The gravimetric method is based on TEOM (tapered element oscillating microbalance) technology, which intakes the atmospheric air through a filter, heats it, continuously measures the filter weight, and calculates the mass concentration of PM in near real time. Therefore, TEOM has high accuracy in field tests compared with

the various air-quality devices. It has been used in many countries to monitor $PM_{2.5}$ concentrations in the field [14].

### 2.1.2. Dataset for Calibration Machine Learning (ML) Modeling

A dataset is required to develop and test the ML model. The dataset includes the labeled time series type by the aforementioned input variables ($PM_{2.5}$ of SPS 30, temperature and humidity of SHT 85) and target variables (TEOM). In general, the dataset is divided into a training set for estimating the ML model parameters and a test set for evaluating the ML model's calibration performance.

For validating the accuracy performance of ML model, the test set should have other combinations of variables that were not utilized in the training set. In addition, the ML model must consider adequate design space between training set and test set. For example, if a variable with a higher concentration range than the training set used for model development is input into the developed ML model, the calibration performance has the possibility of deterioration [15]. In other words, the training set must contain a wide enough range of concentration of $PM_{2.5}$. Additionally, a dataset with a small concentration range of $PM_{2.5}$ may give incorrect evaluations of certain metrics, such as $R^2$ [11]. In this study, low-cost Sensirion SPS30, SHT85 and NRM equipment were collocated to develop the ML model, as shown Figure 2.

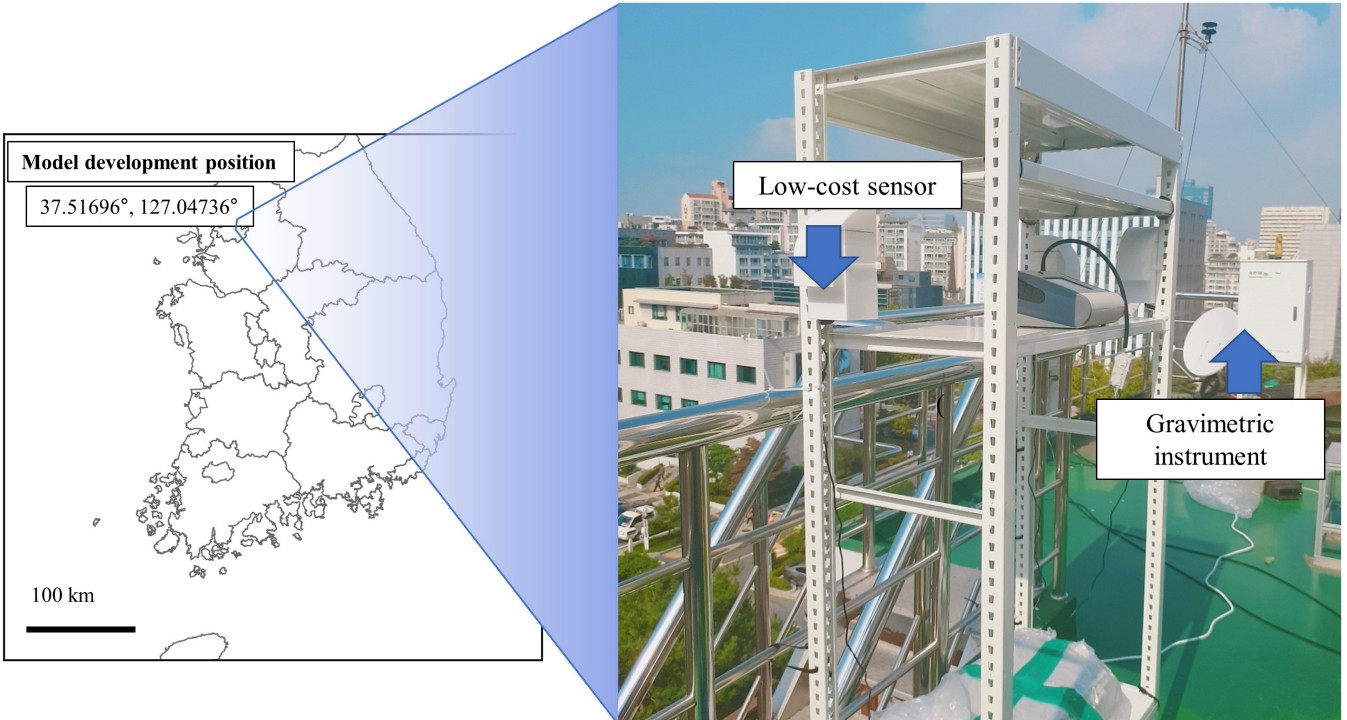

**Figure 2.** Measurement position and field test setup.

### 2.2. Machine Learning Algorithm

The measured $PM_{2.5}$ concentration and environmental variable data (temperature and humidity) have time-series characteristics. That is, air-quality data have a time-dependent characteristic, which is a relationship between past data and current data. Among machine learning algorithms, the long short-term memory (LSTM) neural network is an algorithm optimized for the time-dependent characteristics. In this study, in order to calibrate the low-cost $PM_{2.5}$ sensors, we develop a new $PM_{2.5}$ calibration model by customizing the deep neural network (DNN) optimized in calibration problems and a LSTM optimized in time-dependent characteristics. The overall system architecture of the HybridLSTM algorithm is shown in Figure 3.

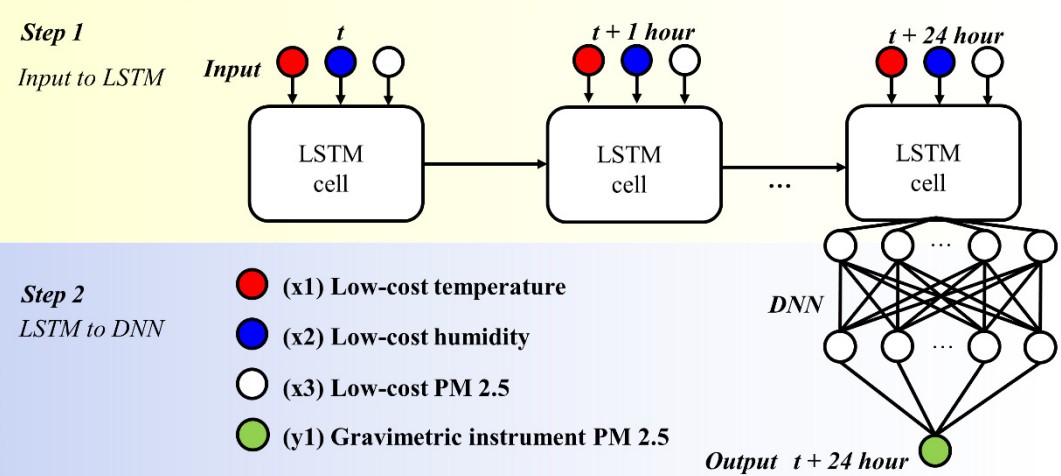

**Figure 3.** Example of HybridLSTM model architecture.

(Step 1) PM$_{2.5}$, temperature and humidity of the low-cost sensor data described in Section 2.1 are input into the network in the form of a time series including historical trends for 24 h. Values entered with historical data provide time-dependent properties between time-series data through LSTM cells.

The LSTM cell computes a non-linear mathematic relation from an input sequence $x = (x_1, \ldots, x_T; x$ is PM$_{2.5}$, temperature and humidity by low-cost sensor) to an output sequence $y = (y_T; y$ is PM$_{2.5}$ by gravimetric instrument) by considering the historical trend using the following equations iteratively from $t = 1$ to $T$ [16,17]:

$$i_t = \sigma(w_{ix}x_t + w_{im}m_{t-1} + w_{ic}c_{t-1} + b_i) \tag{1}$$

$$f_t = \sigma\left(w_{fx}x_t + w_{fm}m_{t-1} + w_{ic}c_{t-1} + b_f\right) \tag{2}$$

$$c_t = f_t \times c_{t-1} + i_t \times g(w_{cx}x_t + w_{cm}m_{t-1} + b_c) \tag{3}$$

$$o_t = \sigma(w_{ox}x_t + w_{om}m_{t-1} + w_{oc}c_t + b_o) \tag{4}$$

$$m_t = o_t \times h(c_t) \tag{5}$$

$$\hat{x}_t = \varphi(w_m m_t + b_x) \tag{6}$$

where $T$ represents the labeled time, $W$ terms denote learning parameter matrices (e.g., $W_{ix}$ is the matrix of weights from the input gate to the inputs), $W_{ic}$, $W_{fc}$, $W_{oc}$ are diagonal learning parameter matrices for peephole connections, the $b$ terms represents bias vectors (bi is the input gate bias vector), $\sigma$ is the sigmoid function, and $i$, $f$, $o$ and $c$ are respectively the input gate, forget gate, output gate and cell activation vectors, all of which are the same size as the cell output activation vector $m$, $\times$ is the element-wise product of the vectors, $h$ and $\varphi$ are tanh and linear activation function, and $\hat{x}_t$ has the new inputs with the historical trend. The predicted vectors are fed into a deep neural network model (DNN).

(Step 2) The time-dependent values are passed to the DNN architecture, and the neural network parameters are trained to minimize the differences between the values of the target variables (TEOM PM$_{2.5}$) and results predicted by the model. The key to the HybridLSTM algorithm is to approach the calibration problem differently from the application of the conventional LSTM approach. For example, the HybridLSTM algorithm is to make the time series of the target variable (TEOM PM$_{2.5}$) the same as the last time series of the input variables (low cost PM$_{2.5}$, temperature and humidity).

The DNN algorithm minimizes the loss between DNN results predicted by the new input design variables ($\hat{x}_t$) and the output variable ($y_T$) by the target data. The structure of the neural network consists of several hidden layers between input and output variables.

The layer consists of various nodes, and the node converts the linear combination of input variables into a sigmoid nonlinear form, as shown in Equations (7) and (8).

$$y_j^{(k)} = b_0 + \sum_{i=1}^{n} w_i x_i \tag{7}$$

$$y_{j\_out}^{(k)} = \frac{1}{1 + \exp\left(-y_j^{(k)}\right)} \tag{8}$$

where $k$ is layer number, $j$ is node number and $w_i$ is weight. The input variables are transferred to the hidden layer and calculated until the end of the output. Then, the weights of all nodes are updated repeatedly so that the error with the true value is minimized. This is called the backpropagation process. That is, parameters such as learning rate, epoch, batch size and number of hidden layers, etc., must be optimized to make the minimum difference value between the true value and prediction value. In other words, HybridLSTM not only has a historical trend for $PM_{2.5}$ by low-cost sensors with humidity and temperature, but also optimizes the loss between results with the historical trend and $PM_{2.5}$ by gravimetric devices as gold standard. In this study, we used Tensorflow and Python 3.6 to model the HybridLSTM.

### *2.3. Benchmark Method*

The multi-linear regression (MLR) method and deep neural network model, which showed high correction performance in previous studies [12], were used as benchmark models to evaluate the performance of the hybridLSTM model proposed in this study. MLR is the same as the equation below;

$$y = \sum_{i=1}^{3} w_i \cdot x_i + b \tag{9}$$

where $x_1$, $x_2$ and $x_3$ are SPS30 $PM_{2.5}$, temperature and humidity, $y$ is the result from TEOM equipment $PM_{2.5}$, and $w$ and $b$ are parameters optimized by the dataset described above.

The DNN model method was explained in Section 2.2. The hyper-parameters of DNN were transferred in previous research [12], which showed reasonable calibration performance. DNN and MLR models are developed using the same training data used to develop the HybridLSTM model, and the model performance is evaluated using the same test data.

The metrics used for model development and evaluation were $R^2$ and root mean square error (RMSE). In general, many performance metrics are used to evaluate regression models, but in evaluating sensor calibration performance, two indicators can be sufficiently explained [11]. The metrics for $R^2$ and RMSE are expressed as follows;

$$R^2 = 1 - \sum_{i=1}^{n} \frac{(t_i - y_i)^2}{(t_i - \bar{t})^2} \tag{10}$$

$$RMSE = \sqrt{\frac{1}{n} \sum_{i=1}^{n} (y_i - \hat{t}_i)^2} \tag{11}$$

where $t_i$, $y_i$, $\bar{t}$ and n represent the $i$-th TEOM sample, the predicted result by the model, the average of the TEOM samples, and the total number of samples, respectively.

## 3. Results

### *3.1. Data Sampling to Develop Machine Learning Model*

The sampled data of time-series type ($PM_{2.5}$ of SPS 30, temperature and humidity of SHT 85 and $PM_{2.5}$ of TEOM) are shown in Figure 4. The data sampling period was measured over 110 days. Seventy-seven days were designated as the training set and

thirty-three days were designated as the test set. The maximum PM$_{2.5}$ concentrations in the training set and test set were sampled for a sufficiently long time to include PM$_{2.5}$ data higher than 50 μg/m$^3$ ($\pm$10 μg/m$^3$) at least, which is the scenario of high concentration determined by the World Health Organization (WHO). Maximum PM$_{2.5}$ concentration measured by the gravimetric method was 115 μg/m$^3$. Therefore, the concentration of PM$_{2.5}$ sampled in this work is high enough to validate the model calibration performance.

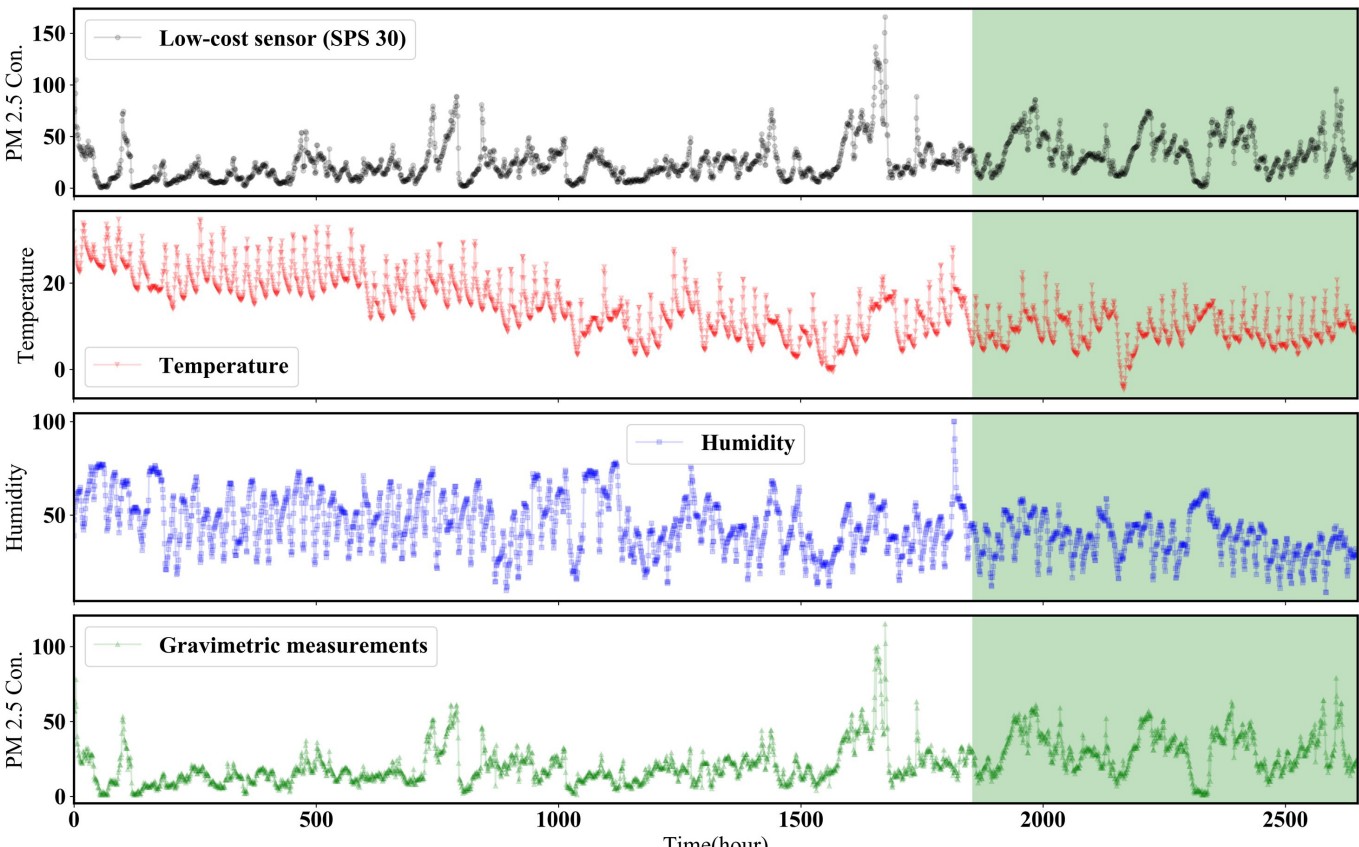

**Figure 4.** Results of collected dataset. From top to bottom, PM$_{2.5}$ of SPS 30, temperature and humidity of SHT 85 and PM$_{2.5}$ of TEOM. The white box region represents the training set that is used to develop the calibration model. The green box region represents the test set that is used to evaluate the developed calibration models. The period of training set is from 30 September 2019 at 18:00 to 18 December 2019 at 16:00. The period of test set is from 18 December 2019 at 17:00 to 21 January 2020 at 18:00. Units of temperature, humidity and PM2.5 concentration are °C, % and μg/m$^3$, respectively.

Table 1 represents the statistical information (maximum, minimum, average and standard deviation for the collected dataset) of the corresponding dataset. We correlate the complexity between input variables (data of low-cost sensors) and target variables (data of high accuracy device) through ML methods based on the data-set.

**Table 1.** Results of statistical information for dataset sampled from low-cost sensor and gravimetric instruments (units of temperature, humidity and PM$_{2.5}$ concentration:°C, % and μg/m$^3$). The dataset is used to validate the accuracy performance of calibrated results.

| Variables | Minimum | Maximum | Average | Standard Deviation |
|---|---|---|---|---|
| Temperature | −4.713 | 34.76 | 13.96 | 6.96 |
| Humidity | 8.55 | 99.99 | 43.51 | 19.45 |
| Low-cost PM$_{2.5}$ | 0.39 | 165.56 | 27.11 | 19.45 |
| Gravimetric PM$_{2.5}$ | 1 | 115 | 22.15 | 14.21 |

### 3.2. Hyper-Parameter Optimization of HybridLSTM

The calibration accuracy by the ML model is affected depending on the combination of hyper-parameters (such as learning rate, network architecture, batch size, optimization function, etc.). Therefore, it is very important to find an optimized hyper-parameter. However, there is a limit to comparing a huge number of combinations. Therefore, many studies determine the hyper-parameter by trial and error methods [18,19]. In this study, hyper-parameters with optimal ML calibration performance were determined by changing various hyper-parameter combinations. Hyper-parameters with optimal calibration performance were evaluated based on $R^2$. Various variables, such as the number of nodes and layers, batch size, etc., were randomly combined into 100 and evaluated, as shown in Figure 5. The $R^2$ for the optimal combination is about 93%, and the hyper-parameter optimization information is summarized in Table 2.

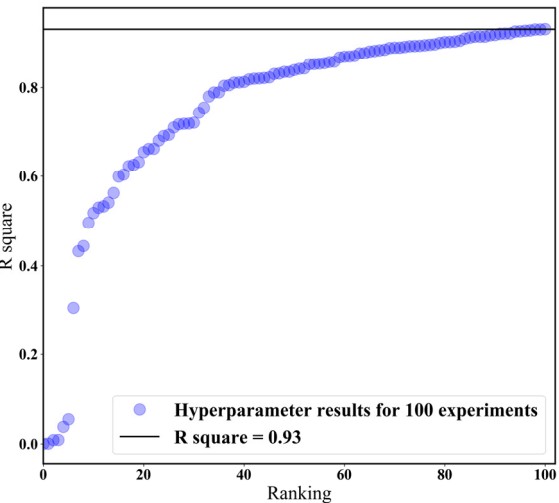

**Figure 5.** Results of hyper-parameter optimization. The experiments were carried out according to 100 experiments. The experiments have different combinations of neural network parameters such as epoch, learning rate and number of layers, etc.

**Table 2.** Results of hyper-parameter optimization for HybridLSTM.

| Optimized Parameters | Values |
| --- | --- |
| Callback | 24 |
| Number of layers | 5 |
| DNN Node | 8/12/24/12/4 |
| Learning rate | 0.0065 |
| Batch size | 15 |
| Epoch | 100 |
| Optimization algorithm | Adam |

Callback is a parameter for how long the LSTM cell gives time-dependency, and the number of DNN layers and nodes are parameters that determine the degree of nonlinearity between the input variable and the output variable. Too many layers cause overfitting and deteriorate the calibration performance of the new input data. The learning rate is that the neural network reduces the loss between the input and output. A learning rate being too large prevents the solution from convergence. Batch size represents the size divided among the entire training set for training the neural network. Epoch refers to the number of iterations to train a neural network. The Adam algorithm was used to optimize the

neural network because the Adam method showed high convergence and accuracy among many algorithms in regression problems [17].

Figure 6 shows the learning process of the HybridLSTM model with the optimized hyper-parameters during 10 training experiments. The validation set determines whether the model have an overfitting problem for new PM$_{2.5}$ data. The training/validation loss was sufficiently converged during the 10 repeated experiments of HybridLSTM, as shown Figure 6. In other words, the developed model represents a robust model without overfitting. Therefore, researchers can develop and use a model with consistent calibration performance.

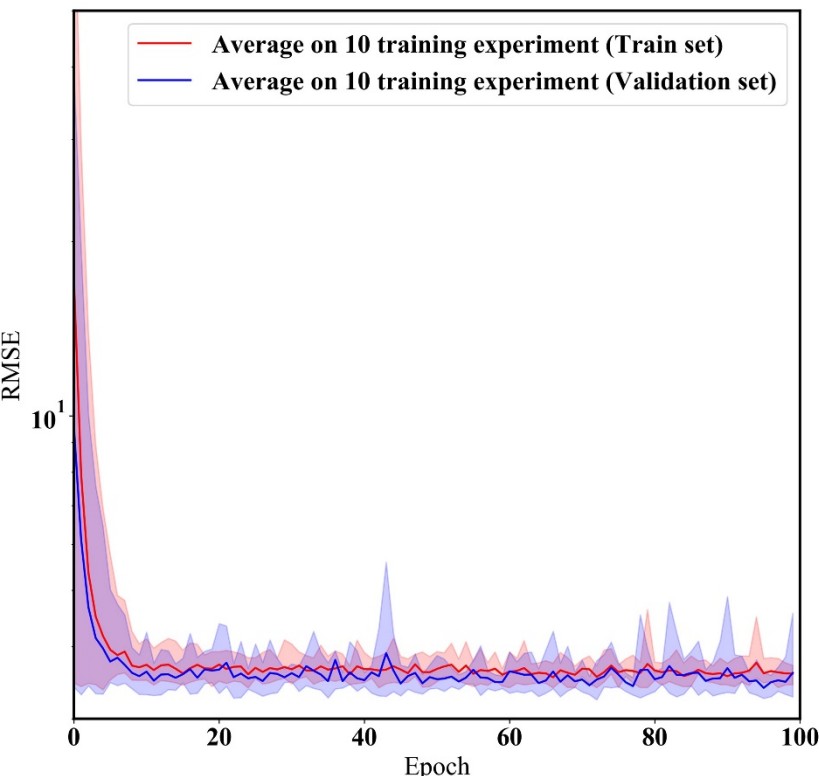

**Figure 6.** Results of robust model test. The robust test is performed on 10 repeated training experiments to validate that the ML model has a consistent calibration performance. The y label is the root-mean-square error (RMSE).

### 3.3. Comparison of Accuracy among Proposed Model, Benchmark and Low-Cost Sensor

Figure 7 shows the RMSE as a result of calibrating the test set at 1-week intervals using the optimized model, benchmark, and SPS30 sensor. The error (RMSE) was calculated based on a gold-standard device (TEOM). The proposed model with time-dependent characteristics showed higher calibration performance for all periods than the benchmark model and raw data. The quantitative error reduction rate for each of the periods is shown in Table 3. The proposed model reduced 41–60% of error compared to the raw data (low-cost sensor), reduced 30–51% of error compared to the calibration results by the MLR model and reduced 8–40% of error compared to the calibration results by the DNN model.

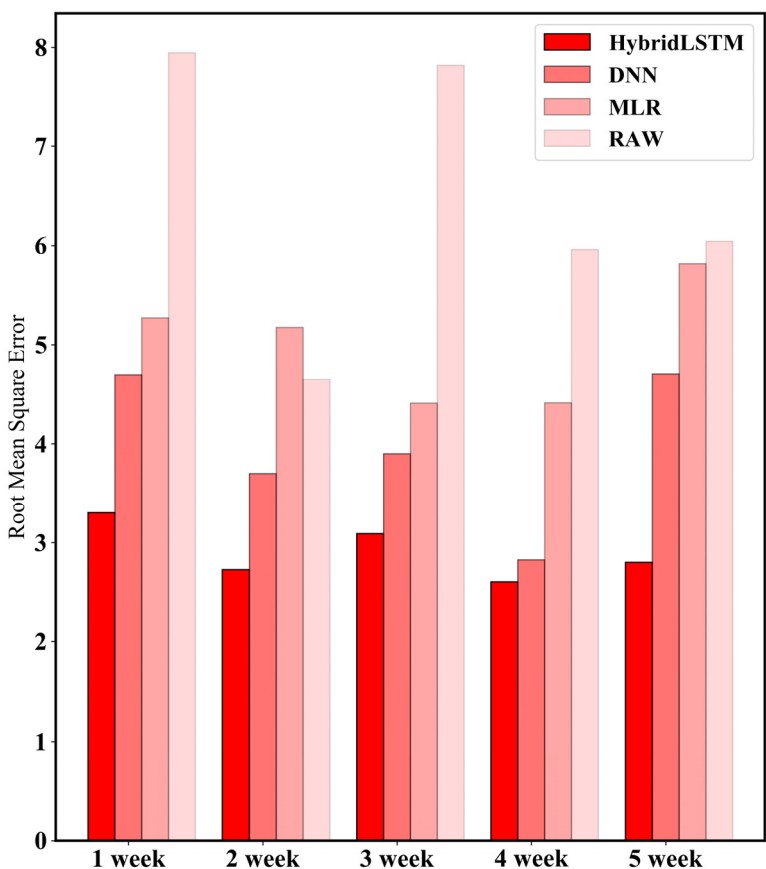

**Figure 7.** Results of error comparison of HybridLSTM, benchmark and raw data at 1-week intervals based on the aforementioned test set. The error (RMSE) was calculated based on gold standard device (TEOM). We tested the calibration performance using datasets collected at weekly intervals. A total of five sections were used (1, 2, 3, 4 and 5 weeks).

**Table 3.** Results of comparison of calibration results of HybridLSTM with benchmark (MLR) and a low-cost sensor (RAW). DNN, MLR and raw data were evaluated based on HybridLSTM. The decrease rate of RMSE from each of models was calculated.

| Decrease Rate of RMSE | 1 Week | 2 Week | 3 Week | 4 Week | 5 Week |
|---|---|---|---|---|---|
| $\frac{\text{DNN} - \text{HybridLSTM}}{\text{DNN}}$ | 29.77% | 26.28% | 20.74% | 7.77% | 40.55% |
| $\frac{\text{MLR} - \text{HybridLSTM}}{\text{MLR}}$ | 37.33% | 47.27% | 29.88% | 43.33% | 50.86% |
| $\frac{\text{RAW} - \text{HybridLSTM}}{\text{RAW}}$ | 58.45% | 41.4% | 60.46% | 58.05% | 52.76% |

Figure 8 represents the comparison results of the developed model, benchmark model, and raw sensor against all samples of TEOM. Raw data by low-cost $PM_{2.5}$ sensors showed a significant overestimation in concentrations higher than 50 $\mu g/m^3$. The incorrect monitoring in high-concentration situations not only leads to incorrect exposure assessment, but also leads to errors in determining government regulations. The benchmark method underestimated compared to the gravimetric method. On the other hand, the proposed model showed small variation results when compared to the TEOM results in the high concentration as well as in the low concentration section. HybridLSTM had the most similar results to TEOM, and the $R^2$ was about 93%. In other words, we showed superior calibration performance of the state-of-the-art machine learning model when compared with the benchmark method (DNN) that is considered to be state of the art in previous literature.

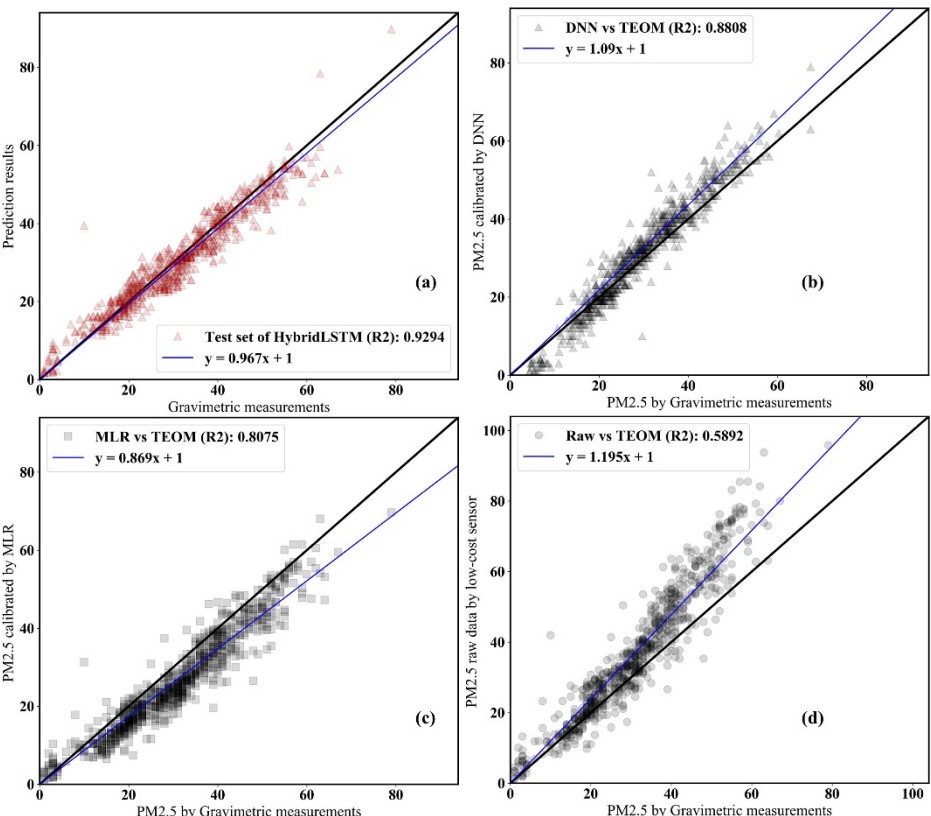

**Figure 8.** Results of scatter plot for the HybridLSTM (**a**), benchmark (**b**) and raw data (**c**) versus gravimetric measurements (**d**).

Figure 9 shows the time-series comparison results of the developed model, benchmark model, and raw sensor against TEOM at 1-week intervals. The dotted line represents the high PM$_{2.5}$ concentrations (>50 μg/m$^3$). The raw data from the low-cost sensor were higher than data from the gravimetric measurement above the dotted line (overestimation). Additionally, the benchmark method represents underestimation compared to the gravimetric instrument under the dotted line. However, the calibration algorithm proposed in this work can calibrate not only the PM$_{2.5}$ of high concentrations, but also the PM$_{2.5}$ of low concentrations in terms of high accuracy.

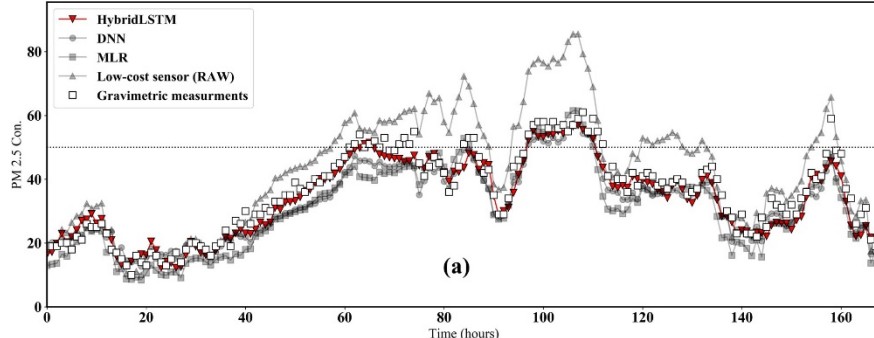

**Figure 9.** *Cont*.

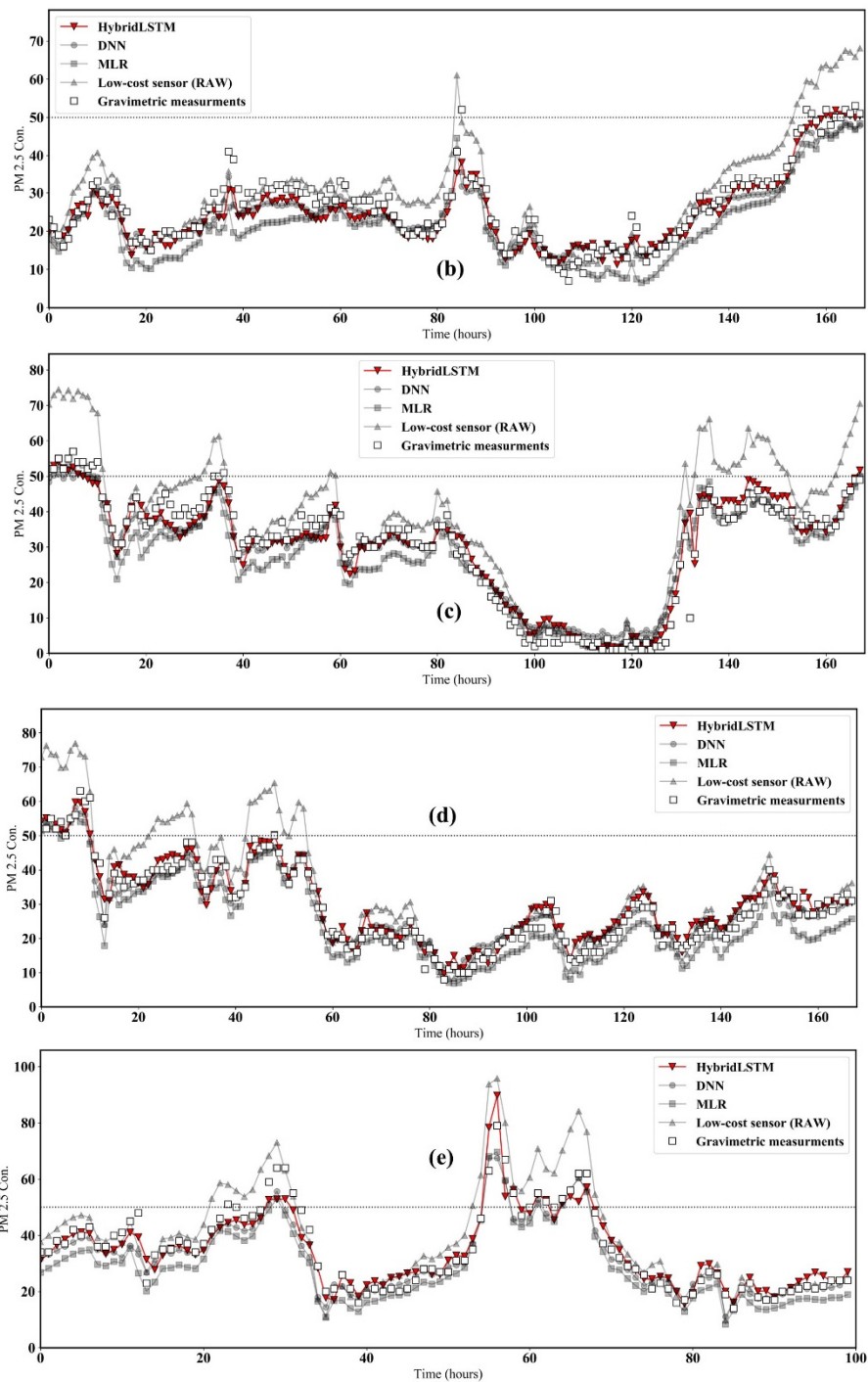

**Figure 9.** Results of time-series comparison results for the HybridLSTM, DNN, MLR and raw data versus gravimetric measurements at 1-week intervals; (**a**) 1, (**b**) 2, (**c**) 3, (**d**) 4 and (**e**) 5 weeks. The dotted line represents the high PM2.5 concentrations. We tested the calibration performance using datasets collected at 1-week intervals. A total of five sections were used (1, 2, 3, 4 and 5 weeks).

## 4. Discussion and Conclusions

In this study, a new $PM_{2.5}$ machine learning calibration model (HybridLSTM) was developed, and the calibration performance was compared with the raw data, MLR model and DNN model, which has shown a superior calibration performance. Additionally, a generalized performance test was performed for validating the possibility of establishing a sensor monitoring network. The results performed are summarized as follows.

(1) The HybridLSTM $PM_{2.5}$ calibration model with time-dependent characteristics showed optimal performance in improving the accuracy of low-cost $PM_{2.5}$ sensors. For RMSE, the proposed model reduced 41–60% of errors compared to the raw data of the low-cost sensor, reduced 30–51% of errors compared to the MLR model and reduced 8–40% of errors compared to the DNN model. Raw data by low-cost PM2.5 sensors showed a significant overestimation compared to the gravimetric method in samples of high $PM_{2.5}$ concentrations. The slope of fitting curve for the raw data and gold standard data was 1.2. The MLR method showed the underestimated calibration results compared to the measurement results by the gravimetric method. HybridLSTM showed a superior calibration performance when compared with ML models (DNN) that are considered to be state of the art in previous literature [12]. HybridLSTM can provide outstanding calibration results for low-cost sensors. Incorrect monitoring in high-concentration situations not only leads to incorrect exposure assessment, but also leads to errors in determining government regulation. The proposed model showed little variation with the NRM method.

The proposed model solves the existing accuracy limitations of low-cost sensors and can provide results with high reliability, not only for monitoring, but also for research in various environmental fields. Although outstanding performance was shown in this study, the method proposed in this study needs to be verified in more locations to build a more reliable sensor monitoring network. Therefore, in future work, we plan to test whether low-cost $PM_{2.5}$ sensors combined with machine learning at various locations and times, including different seasons, can be applied to sensor network construction. When constructing a sensor network with high resolution based on high accuracy, we will test the possibility of providing air quality information to areas where sensors are not installed through the interpolation method.

**Author Contributions:** D.P. was responsible for developing the machine learning algorithm, analyzing the results of calibration by models and oversaw the research and provided initial research ideas. G.-W.Y. researched the machine learning papers, analyzed results from developed machine learning. S.-H.P. collected the dataset and prepared the manuscript. J.-H.L. provided initial research ideas and editing the manuscript. All authors have read and agreed to the published version of the manuscript.

**Funding:** This research was funded by Korea Environment Industry & Technology Institute (KEITI) through Technology Development Project for Safety Management of Household Chemical Products 'Development of aquatic environment exposure index for hazardous substances containing products', funded by Korea Ministry of Environment (MOE) (2020002970009, 1485017560).

**Institutional Review Board Statement:** Not applicable.

**Informed Consent Statement:** Informed consent was obtained from all subjects involved in the study.

**Data Availability Statement:** The data presented in this work are available on request from the corresponding author.

**Conflicts of Interest:** The authors declare no conflict of interest.

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
