# Peer review of "Assessment and Calibration of a Low-Cost PM2.5 Sensor Using Machine Learning (HybridLSTM Neural Network): Feasibility Study to Build an Air Quality Monitoring System"

_atmosphere, doi:10.3390/atmos12101306_

Round 1

Reviewer 1 Report

The authors evaluate the performance of multivariate regression (MLR) and machine learning (ML) methods in calibrating PM 2.5 low-cost sensors, using tepered element oscillating microbalances (TEOMs) as reference instruments. 

The auhtors state that the usage of ML methods vs. MLR benchmarks is a point of novelty for their study, but the reviewer does not agree with this statement. ML methods have been extensively used in low-cost sensors literature to improve sensors' performances. Just to make some examples:

- Wang et al. (2020), for example, has employed both MLR and ML methods (Random Forest and support vector regression) to low-cost PM 2.5 sensors. 

- Similarly, Si et al. (2020) compare ML methods (XGBoost and feedforward neural networks) against low-cost PM sensors. 

- Artificial neural networks were applied to PM 2.5 low-cost sensors also by Bai et al. (2020) which further examined the impact of environmental factors on sensors' performance such as relative humidity. 

Still, the authors never discussed the novelty of their own work in relationship with the known literature making their statement of novelty an unsubstantiated assumption.

Another point of novelty for the authors is the usage of gravimetric methods (specifically TEOMs) as reference methods for low cost PM sensors' calibration, but then again there are multiple examples in literature were gravimetric methods (and TEOMs) were used to different degrees to calibrate low-cost PM sensors (Tryner et al., 2020; Feinberg et al., 2019; Liu et al., 2020; Zusman et al., 2020; Delp & Singer, 2020; Crilley et al., 2020; Seto et al., 2019). Again, the authors do not discuss their claim of novelty in comparison with the existing literature, not making clear in which sense the usage of TEOMs in their work would represent a scientific advancement. Just the paper of Zusman is briefly touched in the introduction, but never again mentioned in the discussion of the results. 

The final point of novelty for the authors is the generalization of the ML model, by employing it on sensors re-located to a different locations. This, again, has been already seen in low-cost sensors' literature. Zimmerman et al. (2018), for example, compares MLR and ML methods (Random Forest, RF) for low-cost gaseous pollutants sensors and evaluate the RF model performance by deploying two low cost sensors in different locations. Again, the author do not discuss the novelty of their generalization in the context of the existing literature and do not clearly highlight what would be the novel aspect of their work. 

The reviewer finds also some lack of information about the dataset employed by the authors. In the manuscript it is stated only the length of the dataset (110 days) and the fact that it was partitioned between training (77 days) and validation (33 days). PM sensors have known to be influenced by environmental parameters such as relative humidity (see for example Bai et al. (2020)) so, which seasons did the dataset span? Did the partition took in account such seasonal factors avoiding to train/validate the model only on seasonal subsets? This concerns the generalization as well: since it is unclear what was the duration of the generalization experiment and when that happened, it should be tested if the good performance of the model holds also in time (i.e.: when the sensors face different seasonal/environmental conditions) beside different spatial position. This is stated in the conclusions, but it is not clearly discussed in the manuscript. 

Finally, the reviewers also thinks that the english used in this manuscript is often unclear and hard to follow, see for example the sentence at lines 247-249 at page 8 ("The proposed model for the loss of the validation set and the train set sufficiently was converged during the repeated training experiment as shown in Figure 5"). The meaning is not clear at all and similarly obscure wording is found throughout the paper.

Overall, taking in account all the previous consideration, the reviewer deems the paper not suitable for publication in its present form. 

Cited Literature:

Bai et al. (2020), “Long-term Field Evaluation of Low-cost Particulate Matter Sensors in Nanjing”.  Aerosol and Air Quality Research, 20, 242–253, doi: 10.4209/aaqr.2018.11.0424

Crilley et al. (2020), “Effect of aerosol composition on the performance of low-cost optical particle counter correction factors”. Atmospheric Measurement Techniques, 13, 1181-1193, doi: 10.5194/amt-13-1181-2020
Delp & singer (2020), “Wildfire Smoke Adjustment Factors for Low-Cost and Professional PM2.5 Monitors with Optical Sensors”. Sensors, 20, 3683, doi: 10.3390/s20133683

Feinberg et al. (2019), “Examining spatiotemporal variability of urban particulate matter and application of high-time resolution data from a network of low-cost air pollution sensors”. Atmospheric Environment 213, 579-584, doi: 10.1016/j.atmosenv.2019.06.026

Liu et al. (2020), “Low-cost sensors as an alternative for long-term air quality monitoring”. Environmental Research, 185, 109438, doi: 10.1016/j.envres.2020.109438

Seto et al. (2020), “Next-Generation Community Air Quality Sensors for Identifying Air Pollution Episodes”. International Research and Public Health, 16, 3268, doi: 10.3390/ijerph16183268

Si et al. (2020), “Evaluation and calibration of a low-cost particle sensor in ambient conditions using machine-learning methods”. Atmospheric Measurement Techniques, 13, 1693–1707, doi: 10.5194/amt-13-1693-2020
Tryner et al. (2020), “Laboratory evaluation of low-cost PurpleAir PM monitors and in-field correction using co-located portable filter samplers”. Atmospheric Environment, 220, 117067, doi: 10.1016/j.atmosenv.2019.117067

Wang et al. (2020), “Application of Machine Learning for the in-Field Correction of a PM2.5 Low-Cost Sensor Network”. Sensors, 20, 5002, doi: 10.3390/s20175002

Zimmerman et al. (2018), “A machine learning calibration model using random forests to improve sensor performance for lower-cost air quality monitoring”. Atmospheric Measurement Techniques, 11, 291–313, doi: 10.5194/amt-11-291-2018

Zusman et al. (2020), “Calibration of low-cost particulate matter sensors: Model development for a multi-city epidemiological study”. Environment International, 134, 105329, doi: 10.1016/j.envint.2019.105329

Author Response

Comments and Suggestions for Authors

The authors evaluate the performance of multivariate regression (MLR) and machine learning (ML) methods in calibrating PM 2.5 low-cost sensors, using tepered element oscillating microbalances (TEOMs) as reference instruments. 

Point 1: The auhtors state that the usage of ML methods vs. MLR benchmarks is a point of novelty for their study, but the reviewer does not agree with this statement. ML methods have been extensively used in low-cost sensors literature to improve sensors' performances. Just to make some examples:

- Wang et al. (2020), for example, has employed both MLR and ML methods (Random Forest and support vector regression) to low-cost PM 2.5 sensors. 

- Similarly, Si et al. (2020) compare ML methods (XGBoost and feedforward neural networks) against low-cost PM sensors. 

- Artificial neural networks were applied to PM 2.5 low-cost sensors also by Bai et al. (2020) which further examined the impact of environmental factors on sensors' performance such as relative humidity. 

Still, the authors never discussed the novelty of their own work in relationship with the known literature making their statement of novelty an unsubstantiated assumption.

Response 1 : According to your suggestion, we introduced that previous studies used machine learning methods to calibrate low-cost sensors. We showed proposed model in this work (HybirdLSTM) has the state-of-arts calibration performance for low-cost PM sensor. That is a point of novelty for our study. Si et al. (2020) showed DNN outperforms the tree-based machine learning model (XGboost), MLR and raw data. To date, DNN has state-of-arts calibration performance in various paper for calibration of low-cost PM 2.5 sensor. Therefore, we compared HybridLSTM with DNN, MLR and raw data. The results of calibration performance of HybridLSTM is as follow that;

Introduction

We added a motivating phrase to our state-of-the-art machine learning algorithms by introducing the citied paper.

“Si et al. [12] introduced machine learning approaches to improve the accuracy problem of the linear regression method for the low-cost sensor calibration. They compared the PM2.5 data calibrated by the simple linear regression (SLR), the multiple linear regression (MLR) the tree-base machine learning algorithm (XGboost) and deep neural network (DNN) against PM2.5 data by the Synchronized Hybrid Ambient Real-time Particulate (SHARP) monitor. They showed the machine learning methods have superior calibration performance compared the linear regression methods. Among the calibration algorithms, DNN showed the best performance for PM2.5 calibrations (person R = 0.85, Root mean square error = 3.91). However, the calibration performance of low-cost sensor can be still improved because various machine learning algorithms can be fused for the purpose of solving them.

In this study, we propose a state-of-arts PM2.5 calibration model (HybridLSTM) by combining the deep neural network (DNN) optimized in calibration problem and a long short-term memory (LSTM) neural network optimized in time-dependent characteristics to improve the performance of conventional calibration algorithms (DNN, MLR) of low-cost PM sensor. This work has two purposes; (1) development on a low-cost calibration machine learning (ML) model and comparison with the previous state-of-arts model (DNN) and conventional MLR model (2) A applicability test to show whether the proposed machine learning algorithm persists the superior performance in the new spatial-temporal field condition.”

Results

We added the results of comparison of HybridLSTM with DNN, MLR and raw data. The gravimetric instrument was used as gold standard. The added phrase is as follow:

“Figure 7 show the RMSE as a result of calibrating the test set at 1-week intervals using the optimized model, benchmark, and SPS-30 sensor. The error (RMSE) was calculated based on gold standard device (TEOM). The proposed model with time-dependent characteristics showed higher calibration performance for all periods than the benchmark model and raw data. The quantitative error reduction rate for each of periods is shown in Table 4. The proposed model reduced 41-60% of error compared to the raw data (low-cost sensor), reduced 30-51% of error compared to the calibration results by MLR model and reduced 8-40% of error compared to the calibration results by DNN model.”

Figure 7. Results of error comparison of HybridLSTM, benchmark and raw data at 1-week intervals based on the aforementioned test set. The error (RMSE) was calculated based on gold standard device (TEOM). We tested the calibration performance using data sets collected at weekly intervals. A total of 5 sections were used (1,2,3,4 and 5 week).

 Table 4. Results of comparison of calibration result of HybridLSTM with benchmark(MLR) and low-cost sensor (RAW). DNN, MLR and raw data were evaluated based on HybridLSTM. The decrease rate of RMSE from each of models were calculated.

Decrease rate of RMSE

1 week

2 week

3 week

4 week

5 week

29.77 %

26.28 %

20.74 %

7.77 %

40.55 %

37.33 %

47.27 %

29.88 %

43.33 %

50.86 %

58.45 %

41.4 %

60.46 %

58.05 %

52.76 %

Figure 8. Results of scatter plot for the HybridLSTM (a), benchmark (b) and raw data (c) versus gravimetric measurements.

Point 2: Another point of novelty for the authors is the usage of gravimetric methods (specifically TEOMs) as reference methods for low cost PM sensors' calibration, but then again there are multiple examples in literature were gravimetric methods (and TEOMs) were used to different degrees to calibrate low-cost PM sensors (Tryner et al., 2020; Feinberg et al., 2019; Liu et al., 2020; Zusman et al., 2020; Delp & Singer, 2020; Crilley et al., 2020; Seto et al., 2019). Again, the authors do not discuss their claim of novelty in comparison with the existing literature, not making clear in which sense the usage of TEOMs in their work would represent a scientific advancement. Just the paper of Zusman is briefly touched in the introduction, but never again mentioned in the discussion of the results.

Response 2: Following your advice, the novelty point of the introduction has been deleted, and our research novelty has been modified as follows.

The added verses are: The novelty of this study is as follow; Improvement of the calibration performance against the existing calibration model (MLR and DNN) by using a new machine learning algorithm. It is very important because it does not only provide reliable community-level monitoring, but also can help exposure assessment in epidemiological studies.

Point 3. The final point of novelty for the authors is the generalization of the ML model, by employing it on sensors re-located to a different locations. This, again, has been already seen in low-cost sensors' literature. Zimmerman et al. (2018), for example, compares MLR and ML methods (Random Forest, RF) for low-cost gaseous pollutants sensors and evaluate the RF model performance by deploying two low cost sensors in different locations. Again, the author do not discuss the novelty of their generalization in the context of the existing literature and do not clearly highlight what would be the novel aspect of their work. 

Response 3: Following your advice, we eliminate the generalization results. We focused the state-of-arts calibration performance results.

Point 4: The reviewer finds also some lack of information about the dataset employed by the authors. In the manuscript it is stated only the length of the dataset (110 days) and the fact that it was partitioned between training (77 days) and validation (33 days). PM sensors have known to be influenced by environmental parameters such as relative humidity (see for example Bai et al. (2020)) so, which seasons did the dataset span? Did the partition took in account such seasonal factors avoiding to train/validate the model only on seasonal subsets? This concerns the generalization as well: since it is unclear what was the duration of the generalization experiment and when that happened, it should be tested if the good performance of the model holds also in time (i.e.: when the sensors face different seasonal/environmental conditions) beside different spatial position. This is stated in the conclusions, but it is not clearly discussed in the manuscript.
Response 4: As your advice, it is true that it is sensitive to the environmental variable humidity. Therefore, in this work, the sampled humidity was sufficiently collected in all measurable ranges from 0 to 99%. In addition, it can be seen that the performance of the low-cost sensor rapidly deteriorates above 50 microgram/m qubic. In this work, sufficient high-concentration fine dust samples were collected in this task.

It is clear that the maintenance of sensor performance over time correlates with generalization.  We collected data samples in various external temperature from -4 to 34 degrees, and the proposed model showed sufficient performance for the test set.

The focus of this study is to develop an algorithm that produces the best performance for low-cost PM sensor calibration in advance. In future studies, considering seasonal factors, data will be accumulated for more than one year and generalization performance will be investigated in more locations.

Point 5: Finally, the reviewers also think that the english used in this manuscript is often unclear and hard to follow, see for example the sentence at lines 247-249 at page 8 ("The proposed model for the loss of the validation set and the train set sufficiently was converged during the repeated training experiment as shown in Figure 5"). The meaning is not clear at all and similarly obscure wording is found throughout the paper.

Response 5: Based on the reviewer's advice, we edited the all typos, errors and incomplete information.

 Overall, taking in account all the previous consideration, the reviewer deems the paper not suitable for publication in its present form. 

Cited Literature:

Bai et al. (2020), “Long-term Field Evaluation of Low-cost Particulate Matter Sensors in Nanjing”.  Aerosol and Air Quality Research, 20, 242–253, doi: 10.4209/aaqr.2018.11.0424

Crilley et al. (2020), “Effect of aerosol composition on the performance of low-cost optical particle counter correction factors”. Atmospheric Measurement Techniques, 13, 1181-1193, doi: 10.5194/amt-13-1181-2020
Delp & singer (2020), “Wildfire Smoke Adjustment Factors for Low-Cost and Professional PM2.5 Monitors with Optical Sensors”. Sensors, 20, 3683, doi: 10.3390/s20133683

Feinberg et al. (2019), “Examining spatiotemporal variability of urban particulate matter and application of high-time resolution data from a network of low-cost air pollution sensors”. Atmospheric Environment 213, 579-584, doi: 10.1016/j.atmosenv.2019.06.026

Liu et al. (2020), “Low-cost sensors as an alternative for long-term air quality monitoring”. Environmental Research, 185, 109438, doi: 10.1016/j.envres.2020.109438

Seto et al. (2020), “Next-Generation Community Air Quality Sensors for Identifying Air Pollution Episodes”. International Research and Public Health, 16, 3268, doi: 10.3390/ijerph16183268

Si et al. (2020), “Evaluation and calibration of a low-cost particle sensor in ambient conditions using machine-learning methods”. Atmospheric Measurement Techniques, 13, 1693–1707, doi: 10.5194/amt-13-1693-2020
Tryner et al. (2020), “Laboratory evaluation of low-cost PurpleAir PM monitors and in-field correction using co-located portable filter samplers”. Atmospheric Environment, 220, 117067, doi: 10.1016/j.atmosenv.2019.117067

Wang et al. (2020), “Application of Machine Learning for the in-Field Correction of a PM2.5 Low-Cost Sensor Network”. Sensors, 20, 5002, doi: 10.3390/s20175002

Zimmerman et al. (2018), “A machine learning calibration model using random forests to improve sensor performance for lower-cost air quality monitoring”. Atmospheric Measurement Techniques, 11, 291–313, doi: 10.5194/amt-11-291-2018

Zusman et al. (2020), “Calibration of low-cost particulate matter sensors: Model development for a multi-city epidemiological study”. Environment International, 134, 105329, doi: 10.1016/j.envint.2019.105329

Reviewer 2 Report

I enjoyed reading through the submitted manuscript. I think this manuscript provides a nice and brief overview of the novel method developed by the authors to calibrate and validate the measurement values reported by the low-cost PM2.5 sensors to a high degree of accuracy. I think the paper is well organized and delivers the content concisely to the readers. The calibration results presented are truly impressive and the use of combined DNN and LSTM optimization hybrid method seems like a clever technique that can be utilized to increase the accuracy of the low-cost sensor readings. It is an interesting topic and similar exercises can be done to other low-cost instruments emerging in the market as well.

Please see the following comments in relation to the marked-up version of the submitted manuscript. 

Major Comments:

  1. I highly recommend getting this manuscript edited by a technical editor for English language accuracy. I found numerous typos and mistakes in the usage of English. I have highlighted in yellow the spots where I detected errors, typos and incomplete information. Please go through each yellow-highlighted item.
  2. In the abstract, it is a little confusing as to what the actual comparison is being made to (benchmark MLR or gravimetric PM5 measurements). Later in the abstract, a sentence says, “Low-cost sensors combined with ML model not only can improve the calibration performance of benchmark, but can also…”. So, it is very confusing as to what is being compared against what here. My understanding is that the low-cost sensor reading (raw) is processed using the ML algorithm to produce a calibrated dataset, which is then compared against (a) benchmark MLR model, and (b) gravimetric measurement as the reference datum, correct? If so, the abstract needs to be re-written to make it sound less confusing.
  3. Since this paper mainly deals with the calibration of measured values, it is also important to note the accuracy/uncertainty of each measurement along with the reported values (E.g., on line 146: 50 µg/m3 ± x µg/m3, where x is the uncertainty associated with the measurement).
  4. Please include references to provide picture credit and make sure that you have received permissions from the manufacturers before using images of the instruments in Figure 2. Please note that the images may be subject to copyright protection.
  5. Table 2 shows the hyper-parameters finally chosen by the authors based on trial and error. It would complement this table to show with a graph the actual “optimization” that happened as the different values of the hyper-parameters were experimented with. Also, what was the optimized parameter? The RMSE between the gravimetric reading and the model-predicted value? It is worth mentioning here to convince the reader that the hyper-parameters listed in Table 2 are indeed the optimal values.
  6. More description of the Hybrid LSTM model itself would be helpful to the reader willing to replicate the study and investigate the accuracy of your claims. Parameters that would fully define the model (along with the units of inputs and outputs), a schematic diagram of the detailed model etc. would be useful to provide detailed description of the ML model. I also encourage you to share your data and model and/or associated codes as supplementary materials to make your paper more transparent.

Minor/Technical Comments:

  1. The text embedded in Figures 1 and 7 are too small in font size. It can be illegible to some readers. Please increase the font size.
  2. Line 109: Please provide relevant quantitative data from the cited reference to justify that the “SPS 30 has excellent inter-sensor precision”.
  3. Please indicate the units of measurement in the Y-axes of each sub-figure in Figure 3 as well as in Table 1.
  4. Lines 284-285: Please clarify if the “high PM5 concentrations” mean that they are above a certain regulatory threshold value. If so, please mention the value, in what country, and provide relevant reference to the regulatory statue.

Author Response

Comments and Suggestions for Authors

I enjoyed reading through the submitted manuscript. I think this manuscript provides a nice and brief overview of the novel method developed by the authors to calibrate and validate the measurement values reported by the low-cost PM2.5 sensors to a high degree of accuracy. I think the paper is well organized and delivers the content concisely to the readers. The calibration results presented are truly impressive and the use of combined DNN and LSTM optimization hybrid method seems like a clever technique that can be utilized to increase the accuracy of the low-cost sensor readings. It is an interesting topic and similar exercises can be done to other low-cost instruments emerging in the market as well.

Please see the following comments in relation to the marked-up version of the submitted manuscript. 

Major Comments:

Point 1: I highly recommend getting this manuscript edited by a technical editor for English language accuracy. I found numerous typos and mistakes in the usage of English. I have highlighted in yellow the spots where I detected errors, typos and incomplete information. Please go through each yellow-highlighted item.

Response 1: Based on the reviewer's advice, we edited the all typos, errors and incomplete information that you recommend the yellow-highlighted item.

    Point 2: In the abstract, it is a little confusing as to what the actual comparison is being made to (benchmark MLR or gravimetric PM2.5 measurements). Later in the abstract, a sentence says, “Low-cost sensors combined with ML model not only can improve the calibration performance of benchmark, but can also…”. So, it is very confusing as to what is being compared against what here. My understanding is that the low-cost sensor reading (raw) is processed using the ML algorithm to produce a calibrated dataset, which is then compared against (a) benchmark MLR model, and (b) gravimetric measurement as the reference datum, correct? If so, the abstract needs to be re-written to make it sound less confusing.

Response 2 : Based on the reviewer's advice, we added the follow sentence.

“The proposed model was compared with benchmarks (multiple linear regression model, deep neural network model) and low-cost sensor results. The gravimetric measurements were used as reference data to evaluate sensor accuracy”

“In other words, the proposed ML model has the state-of-arts calibration performance among the calibration algorithms.”

Point 3: Since this paper mainly deals with the calibration of measured values, it is also important to note the accuracy/uncertainty of each measurement along with the reported values (E.g., on line 146: 50 µg/m3 ± µg/m3, where x is the uncertainty associated with the measurement).

Response 3: Based on the reviewer's advice, we edited the sentence to consider the uncertainty of low-cost sensor (SPS30).
line 146: 50

Point 4: Please include references to provide picture credit and make sure that you have received permissions from the manufacturers before using images of the instruments in Figure 2. Please note that the images may be subject to copyright protection.

Response 4:At your review point, the picture has been removed under copyright.

Point 5: Table 2 shows the hyper-parameters finally chosen by the authors based on trial and error. It would complement this table to show with a graph the actual “optimization” that happened as the different values of the hyper-parameters were experimented with.

Also, what was the optimized parameter? The RMSE between the gravimetric reading and the model-predicted value? It is worth mentioning here to convince the reader that the hyper-parameters listed in Table 2 are indeed the optimal values.

Response 5: According to your review, R2 was set as a reference value to find the hyper-parameters with optimal model calibration performance. The figure that derives the highest R2 according to 100 random hyper-parameter experiments was added as follows.
The added sentence is as follow that;
“In this study, hyper-parameters with optimal ML calibration performance were determined by changing various hyper-parameter combinations. Hyper-parameters with optimal calibration performance were evaluated based on R2. Various variables such as the number of nodes and layers, batch size, etc. were randomly combined into 100 and evaluated as shown Fig. The R2 for the optimal combination is about 93%, and the hyper-parameter optimization information are summarized in Table 2.”

Point 6: More description of the Hybrid LSTM model itself would be helpful to the reader willing to replicate the study and investigate the accuracy of your claims. Parameters that would fully define the model (along with the units of inputs and outputs), a schematic diagram of the detailed model etc. would be useful to provide detailed description of the ML model. I also encourage you to share your data and model and/or associated codes as supplementary materials to make your paper more transparent.

Response 6: For easier understanding of the reader, the algorithm process has been summarized using a formula. Also, data and code will be provided in person when the reader requests it.

(Step 1) PM2.5, temperature, and humidity of the low-cost sensor data described in Section 2.1 are input to the network in the form of a time series including historical trends for 24 hours. Values entered with historical data provide time-dependent properties between time series data through LSTM cells.

LSTM cell computes a non-linear mathematic relation from an input sequence x = (x1, ..., xT; x is PM2.5, temperature and humidity by low-cost sensor) to an output sequence y = (yT; y is PM2.5 by gravimetric instrument) by considering the historical trend using the following equations iteratively from t = 1 to T [15,16]:

(1)

(2)

(3)

(4)

(5)

(6)

where the T represents the labeled time, the W terms denote learning parameter matrices (e.g. Wix is the matrix of weights from the input gate to the inputs), Wic, Wfc, Woc are diagonal learning parameter matrices for peephole connections, the b terms represents bias vectors (bi is the input gate bias vector), σ is the sigmoid function, and i, f, o and c are respectively the input gate, forget gate, output gate and cell activation vectors, all of which are the same size as the cell output activation vector m,  is the element-wise product of the vectors, h and φ are tanh and linear activation function, t has the new inputs with the historical trend. The predicted vectors are fed into deep neural network model (DNN).

(Step 2) The time-dependent values are passed to the DNN architecture, and the neural network parameters are trained to minimize the differences between the values of the target variables (TEOM PM2.5) and results predicted by the model. The key to the HybridLSTM algorithm is to approach the calibration problem differently from the application of conventional LSTM approach. For example, HybridLSTM algorithm is to make the time series of the target variable (TEOM PM2.5) the same as the last time series of the input variables (low cost PM2.5, temperature and humidity).

The DNN algorithm minimize the loss between DNN results predicted by the new input design variables (t) and the output variable (yT) by the target data. The structure of the neural network consists of several hidden layers between input and output variables. The layer consists of various nodes, and the node converts the linear combination of input variables into a sigmoid nonlinear form as shown in Equation (7) and Equation (8).

(7)

(8)

Where  is layer number,  is node number and  is weight. The input variables are transferred to the hidden layer and calculated until the end of the output. Then, the weight of all nodes are updated repeatedly so that the error with the true value is minimized. This is called backpropagation process. That is, the parameters such as learning rate, epoch, batch size and number of hidden layers etc. must be optimized to make the minimum difference value between the true value and prediction value. In other words, HybridLSTM not only has the historical trend for PM2.5 by low-cost sensor with humidity and temperature, but also optimize the loss between results with the historical trend and PM2.5 by gravimetric device as gold standard. In this study, we used Tensorflow and Python 3.6 to model the HybridLSTM.

Minor/Technical Comments:

Point 7: The text embedded in Figures 1 and 7 are too small in font size. It can be illegible to some readers. Please increase the font size.

Response 7: According to your suggestion, we increase the font size in Figure 1 and 7

<edited figure 1>

<edited figure 7>

Point 8: Line 109: Please provide relevant quantitative data from the cited reference to justify that the “SPS 30 has excellent inter-sensor precision”.

Response 8: According to your suggestion, we added a sentence
“It is very important to have consistent precision among low-cost sensors in order to build a monitoring sensor network system by ML model. Because the low-cost Sensirion SPS 30 has excellent inter-sensor precision with coefficients of determination above 0.9 [10]”

Point 9: Please indicate the units of measurement in the Y-axes of each sub-figure in Figure 3 as well as in Table 1.

Response 9: According to your suggestion, we indicated the unit of measurement in figure and table cation.

Point 10: Reviewer 2 Lines 284-285: Please clarify if the “high PM5 concentrations” mean that they are above a certain regulatory threshold value. If so, please mention the value, in what country, and provide relevant reference to the regulatory statue.

Response 10: The high PM2.5 concentrations line was used to represent the overestimation of raw data by low-cost sensor against the gravimetric results. The overestimation was showed above the line, we added this sentence “ The dot line represents the high PM2.5 concentrations (> 50 g/m3). The raw data from low-cost sensor showed the higher than data from the gravimetric measurement above the dot line (overestimation). Also, the benchmark method represents the underestimation compared to the gravimetric instrument under dot line.”

Reviewer 3 Report

The authors developed a hybrid-LSTM neural network to calibrate PM2.5 measurements collected using low-cost sensors air pollution sensors. The authors compared the machine learning calibration model to the traditionally used multiple linear regression model and found better performance of the machine learning calibration method. Additionally, the authors collected PM2.5 samples at two different locations to test the generalizability of the machine learning methods. Overall, I found the methods used in the study sound and the work is important to the exposure science society. The manuscript might be improved in the following aspects:

Abstract

Line 10: the low-cost sensors has not been used for regulation purpose. The sentence should be revised.

Introduction:

Line 32-34: citations are needed to support this information.

Line 52-55: the two sentences were contradictory with each other. I guess the authors want to show that the SPS 30 sensors has higher accuracy as compared to other low-cost sensor, but still lower accuracy than the NRF measurements.

Line 65-66: While the authors introduced that previous studies used multiple linear regression to calibrate low-cost sensors, the motivation to use machine leaning algorithm is not well stated. For example, why the authors think ML might performed better than MLR? More rationale and motivations should be included in the introduction.

Methods

The methods and results section were mixed. Some of the results were included into the methods section (e.g., Table 1, Table 2). Also some paragraph in the results section should be moved to the methods section ( see details in the follow comments).

Method 2.1.2: The authors used the first 77 days as the training dataset and the later 33 days as the testing dataset. While most modeling work divide the whole dataset into 10 folds randomly, using 9 of them as the training set and the rest one fold as the testing set. IIs there specific reasons that the authors did not separate the dataset randomly?

Figure 4: it might be better to reverse the figure so that step 1 is above of step 2.

Method 2.2: line 205-207, which metric was used to determine the performance of models with varying hyper-parameters?

Results

The descriptive analysis should be the first part of the results section. I suggest the authors move Table 1 to the start of the results section. Also, a scatter plot of low-cost sensor measured PM2.5 vs. NRF measured PM2.5 would be super helpful for readers to understand the data.

Figure 6 and 8:  it is not clear what does the 1week-5week mean in the figures. The authors can describe the results a little bit in the caption.

Results 3.2: line 302-307 should be introduced in the methods section.

Throughout the manuscript, the authors should check and correct all the typos and grammatical mistakes.

Author Response

Comments and Suggestions for Authors

The authors developed a hybrid-LSTM neural network to calibrate PM2.5 measurements collected using low-cost sensors air pollution sensors. The authors compared the machine learning calibration model to the traditionally used multiple linear regression model and found better performance of the machine learning calibration method. Additionally, the authors collected PM2.5 samples at two different locations to test the generalizability of the machine learning methods. Overall, I found the methods used in the study sound and the work is important to the exposure science society. The manuscript might be improved in the following aspects:

Abstract

Point 1: Line 10: the low-cost sensors has not been used for regulation purpose. The sentence should be revised.
Response 1: According to your suggestion, we edited the sentence as follow that;
“Commercially-available low-cost air quality sensors have low accuracy. If the low accuracy problem is solved, the low-cost sensor system has possibility to investigate rationally PM2.5 emission caused by industrial activities or to estimate accurately the personal exposure for PM2.5.”

Introduction:

Point 2: Line 32-34: citations are needed to support this information.
Response 2: we added the reference to support the information.

Conti, M.E.; Ciasullo, R.; Tudino, M.B.; Matta, E.J. The industrial emissions trend and the problem of the implementation of the Industrial Emissions Directive (IED). Air Qual. Atmos. Heal. 2015, 8, 151–161, doi:10.1007/s11869-014-0282-7.

Point 3: Line 52-55: the two sentences were contradictory with each other. I guess the authors want to show that the SPS 30 sensors has higher accuracy as compared to other low-cost sensor, but still lower accuracy than the NRF measurements.

Response 3: In order to clarify the context, we edited the sentence as follow;

The SPS 30 sensor has higher accuracy and high correlation as compared to other low-cost sensors. However, it has still been shown that the low-cost sensor has lower accuracy than the national standard measurements due to the limitations of the physical characteristics of the sensor.

Point 4: Line 65-66: While the authors introduced that previous studies used multiple linear regression to calibrate low-cost sensors, the motivation to use machine leaning algorithm is not well stated. For example, why the authors think ML might performed better than MLR? More rationale and motivations should be included in the introduction

Response 4: According to your review point, we added the sentence to explain the motivation of using the machine learning algorithm.
“Si et al. [12] introduced machine learning approaches to improve the accuracy problem of the linear regression method for the low-cost sensor calibration. They compared the PM2.5 data calibrated by the simple linear regression (SLR), the multiple linear regression (MLR) the tree-base machine learning algorithm (XGboost) and deep neural network (DNN) against PM2.5 data by the Synchronized Hybrid Ambient Real-time Particulate (SHARP) monitor. They showed the machine learning methods have superior calibration performance compared the linear regression methods. Among the calibration algorithms, DNN showed the best performance for PM2.5 calibrations (person R = 0.85, Root mean square error = 3.91). However, the calibration performance of low-cost sensor can be still improved because various machine learning algorithms can be fused for the purpose of solving them.”

Methods

Point 5: The methods and results section were mixed. Some of the results were included into the methods section (e.g., Table 1, Table 2). Also some paragraph in the results section should be moved to the methods section (see details in the follow comments).

Response 5: According to your review point, we included the Table 1 into results section. Table 2 is the results of algorithm development, so it was moved to the results.

Point 6: Method 2.1.2: The authors used the first 77 days as the training dataset and the later 33 days as the testing dataset. While most modeling work divide the whole dataset into 10 folds randomly, using 9 of them as the training set and the rest one fold as the testing set. IIs there specific reasons that the authors did not separate the dataset randomly?

Response 6: The k-fold method is generally used when the machine learning model performance is not good due to insufficient data. In this study, the number of data samples collected are sufficient to validate the machine learning model performance. In addition, the proposed LSTM algorithm requires continuous time series data. In k-fold method occurs non-continuous time series data samples. Therefore, we divided the data set by about 7:3 and tested it, and showed sufficient correction performance.

Point 7: Figure 4: it might be better to reverse the figure so that step 1 is above of step 2.

Response 7: According to your review, steps 1 and 2, which are the order of the figure, have been reversed.

Point 8: Method 2.2: line 205-207, which metric was used to determine the performance of models with varying hyper-parameters?

Response 8: According to your review, R2 was set as a reference value to find the hyper-parameters with optimal model calibration performance. The figure that derives the highest R2 according to 100 random hyper-parameter experiments was added as follows.
The added sentence is as follow that;
“In this study, hyper-parameters with optimal ML calibration performance were determined by changing various hyper-parameter combinations. Hyper-parameters with optimal calibration performance were evaluated based on R2. Various variables such as the number of nodes and layers, batch size, etc. were randomly combined into 100 and evaluated as shown Fig. The R2 for the optimal combination is about 93%, and the hyper-parameter optimization information are summarized in Table 2.”

Results

Point 9: The descriptive analysis should be the first part of the results section. I suggest the authors move Table 1 to the start of the results section. Also, a scatter plot of low-cost sensor measured PM2.5 vs. NRF measured PM2.5 would be super helpful for readers to understand the data.

Response 9: The analysis result for dataset was moved in results section according to your suggestion. Also, we edited legend of the scatter plot

Point 10: Figure 6 and 8:  it is not clear what does the 1week-5week mean in the figures. The authors can describe the results a little bit in the caption.

Response 10: According to your review point, we added the description in Figure 7 as follow;

“We tested the calibration performance using data sets collected at 1-week intervals. A total of 5 sections were used (1,2,3,4 and 5 week).”

Point 11: Results 3.2: line 302-307 should be introduced in the methods section.

Response 11: According to your review point, we moved the sentence in method section.

Point 12: Throughout the manuscript, the authors should check and correct all the typos and grammatical mistakes.

Response 1: Based on the reviewer's advice, we edited the all typos, errors and incomplete information.

Round 2

Reviewer 1 Report

The authors have addressed the comments proposed in the past review and, while something has bettered there are still some major modifications to be made before the paper is acceptable for publication.

The English still need major revision, not so much in the typos, but rather in the formulation of the sentences themselves. For example, in the Conclusion paragraph, the authors state that “HybridLSTM showed the superior calibration performance compared with DNN that has been showed the state-of-arts calibration performance among machine learning model”. That’s not clear English at all. I guess that what the authors meant is that HybridLSTM proved to be superior compared to ML models that are considered state of the art: “HybridLSTM showed a superior calibration performance when compared with ML models (DNN) that are considered to be state of the arts by previous literature”. Many sentences like this one are found throughout the paper and the reviewer strongly suggest that the authors have the manuscript revised by a native English speaker for clarity. Also, please correct all “state-of-the-arts” in “state-of-the-art”.

The authors have clarified the novelty compared to the previous version of the manuscript. Their novelty would be the application of a combined (hybrid) ML method that yield a better performance in calibrating PM 2.5 sensors rather that what was found in literature, so much that the authors state at the end of the abstract that “In other words, the proposed ML model has the state-of-arts calibration performance among the calibration algorithms”. In the reviewer’s view that sentence should be corrected with “among the –tested- calibration algorithms”, given that in the paper the HybridLSTM method has been compared only against DNN and MLR. Are there are no other papers that have employed HybridLSTM on PM 2.5 data? Or is this the first one? If it is, I would make this much clearer and underline this in the introduction and discussion. If it is not, then the novelty should be better discussed.

In their response, the authors have clarified that during data sampling a full range of temperatures and relative humidities have been sampled by the sensors. Nevertheless it would be much clearer to show the actual periods and season sampled by the sensors substituting, in figure 4 the Time (hour) in the x-axis with a UTC date-time value (e.g.: “1-Jun-2020 07:00”). That would made much clearer how the sampled dataset is divided between training and validation sets.

A final note: in the new pdf version of the manuscript provided, many of the comments are cut. In the .docx version of the manuscript would have been possible to expand the comment bubbles by clicking on the 3 dots on them, but in the pdf version this is not feasible (the comment bubble is like an image). In the next version I suggest to find a way to better visualize comments to give full readability to them.

Overall the reviewer finds that the authors have better focused their manuscript compared to the previous version, but there are still things to adjust, especially considering the clarity of the language of the manuscript

Author Response

Point 1: The English still need major revision, not so much in the typos, but rather in the formulation of the sentences themselves. For example, in the Conclusion paragraph, the authors state that “HybridLSTM showed the superior calibration performance compared with DNN that has been showed the state-of-arts calibration performance among machine learning model”. That’s not clear English at all. I guess that what the authors meant is that HybridLSTM proved to be superior compared to ML models that are considered state of the art: “HybridLSTM showed a superior calibration performance when compared with ML models (DNN) that are considered to be state of the arts by previous literature”. Many sentences like this one are found throughout the paper and the reviewer strongly suggest that the authors have the manuscript revised by a native English speaker for clarity. Also, please correct all “state-of-the-arts” in “state-of-the-art”.

Response 1: Following your advice, we edited many formulation of the sentences as follow. If more English proofing is required, I will use the MDPI proofreading service after accepting the paper.

In abstract

Before change: “If the low accuracy problem is solved, the low-cost sensor system has possibility to investigate rationally PM2.5 emission caused by industrial activities or to estimate accurately the personal exposure for PM2.5.”à
After change: “The improved accuracy of low-cost PM2.5 sensors allows the use of low-cost sensor systems to reasonably investigate PM2.5 emissions from industrial activities or to accurately estimate individual exposure to PM2.5.”

Before change: “For root mean square error (RMSE) for PM2.5 concentrations, the proposed model reduced 41-60% of error compared to the raw data of low-cost sensor, reduced 30-51% of error compared to the MLR model and reduced 8-40% of error compared to the MLR model.”à

After Change: “For root mean square error (RMSE) for PM2.5 concentrations, the proposed model reduced 41-60% of error when compared with the raw data of low-cost sensor, reduced 30-51% of error when compared with the MLR model and reduced 8-40% of error when compared with the MLR model.” (Line 8-10)

Before change: R2 of HybridLSTM, DNN, MLR and raw data were 93, 90, 80 and 59 %.à

After change: R2 of HybridLSTM, DNN, MLR and raw data were 93, 90, 80 and 59 %, respectively. (Line 18-21)

Before change: state-of-artsà

After change: state-of-the-art (Line 22)

Introduction

Before change: installing NRM equipment at each of short-distance sampling locations is expensive à

After change: It is expensive to install NRM equipments at the sampling location for each close distance. (Line 39-41)

After change: “Light-scattering low-cost PM2.5 sensors is paradigm to –“ (Line 41)

Method

Before change: Therefore, the PM2.5 measurement results by the SPS 30 sensor is set as the input variable. à

After change: Therefore, the PM2.5 measurement results by the SPS 30 sensor is set as the input variable based on previous literature [10]. (Line 112-113)

Before change: Dataset includes the labeled time series type by the aforementioned input variables (PM2.5 of SPS 30, temperature and humidity of SHT 85) and target variables (TEOM). à

After change: Dataset consists of the aforementioned input variables (PM2.5 of SPS 30, temperature and humidity of SHT 85) and target variables (TEOM). (Line 128-130)

Before change: the test set must have different combinations of variables compared with the train set. à

After change: the test set should have other combinations of variables that were not utilized in the training set. (Line 133-134)

Results

Before change: In other words, we showed superior calibration performance of the state-of-the-art machine learning model compared with the existing benchmark method (DNN). à

After change: In other words, we showed superior calibration performance of the state-of-the-art machine learning model whencompared with the benchmark method (DNN) that that are considered to be state of the arts by previous literature. (Line 318-321)

In Conclusion

Before change: “HybridLSTM showed the superior calibration performance compared with DNN that has been showed the state-of-the-art calibration performance among machine learning model.” à

After change: “HybridLSTM showed a superior calibration performance when compared with ML models (DNN) that are considered to be state of the arts by previous literature” (Line 354-356)

Point 2-1: The authors have clarified the novelty compared to the previous version of the manuscript. Their novelty would be the application of a combined (hybrid) ML method that yield a better performance in calibrating PM 2.5 sensors rather that what was found in literature, so much that the authors state at the end of the abstract that “In other words, the proposed ML model has the state-of-arts calibration performance among the calibration algorithms”. In the reviewer’s view that sentence should be corrected with “among the –tested- calibration algorithms”, given that in the paper the HybridLSTM method has been compared only against DNN and MLR.

Response 2-1: The sentence was edited based on your advice.

Before change: In other words, the proposed ML model has the state-of-the-art calibration performance among calibration algorithms.

After change: In other words, the proposed ML model has the state-of-the-art calibration performance among the tested calibration algorithms.

Point 2-2: Are there are no other papers that have employed HybridLSTM on PM 2.5 data? Or is this the first one? If it is, I would make this much clearer and underline this in the introduction and discussion. If it is not, then the novelty should be better discussed.

Response 2-2: Also, following your advice, we underline the originality on using HybridLSTM for PM 2.5 data in the introduction. The added sentence is as follow;

 “As far as we can tell, this paper is the first report to propose the new approach of calibration of low-cost PM2.5 sensor by using HybridLSTM algorithm.” (Line 98-100)

Point 3: In their response, the authors have clarified that during data sampling a full range of temperatures and relative humidities have been sampled by the sensors. Nevertheless it would be much clearer to show the actual periods and season sampled by the sensors substituting, in figure 4 the Time (hour) in the x-axis with a UTC date-time value (e.g.: “1-Jun-2020 07:00”). That would made much clearer how the sampled dataset is divided between training and validation sets.

Response 3: We added the caption in Figure 4. The added sentence in caption of figure is as follow;

Figure 4. Results of collected data set; From top to bottom, PM2.5 of SPS 30, temperature and humidity of SHT 85 and PM2.5 of TEOM. Region of white box represents the train set that is used to develop the calibration model. Region of green box represents the test set that is used to evaluate the developed calibration models. The period of training set is from 30-Sep-2019 18:00 to 18-Dec-2019 16:00. The period of test set is from 18-Dec-2019 17:00 to 21-Jan-2020 18:00. Unit of temperature, humidity and PM2.5 concentration are , % and , respectively.

Point 4: A final note: in the new pdf version of the manuscript provided, many of the comments are cut. In the .docx version of the manuscript would have been possible to expand the comment bubbles by clicking on the 3 dots on them, but in the pdf version this is not feasible (the comment bubble is like an image). In the next version I suggest to find a way to better visualize comments to give full readability to them.

Response 4: We submit the zip file that contains this response file and manuscript file. Thanks!

Overall the reviewer finds that the authors have better focused their manuscript compared to the previous version, but there are still things to adjust, especially considering the clarity of the language of the manuscript
